# Impact of Different Types of Lipids on the Quality of Frozen Dough and Bread

**DOI:** 10.3390/foods14234032

**Published:** 2025-11-25

**Authors:** Rui Gao, Kai Yan, Jian Xia, Zixuan Yang, Zhan Wang

**Affiliations:** 1College of Food Science and Engineering, Wuhan Polytechnic University, Wuhan 430048, China; g3158709753@163.com (R.G.); 15549689023@163.com (K.Y.); xja0501@whpu.edu.cn (J.X.); 2Key Laboratory for Deep Processing of Major Grain and Oil, Ministry of Education, Wuhan 430023, China

**Keywords:** lipid oils, solid fats, freeze–thaw cycles, baking performance

## Abstract

This study investigated the impacts of different lipids (liquid oils: corn, peanut, soybean, rapeseed; solid fats: butter, shortening, margarine, lard, coconut oil) on the quality of frozen dough and bread. By comparing F0 (no freeze–thaw) and F2 (two freeze–thaw cycles), the impacts on dough texture, rheology, water distribution, differential scanning calorimetry (DSC), infrared analysis, microstructure, and baking performance were assessed. After F2, corn oil and peanut oil mitigated the increase in hardness. Solid fats better preserved dough viscoelasticity and bound water, thereby minimizing ice recrystallization and structural damage and achieving higher sensory scores, whereas liquid oils reduced the hardness of frozen bread and increased specific volume. Overall, liquid oils and solid fats displayed complementary advantages. This study offers innovative insights and practical value for the frozen-bakery food industry.

## 1. Introduction

As a widely consumed staple, bread often experiences quality deterioration during storage. To overcome this, frozen dough technology, originating in Europe [1], has become essential in modern baking. However, repeated freeze–thaw cycles lead to decreased bread volume, increased firmness, and flavor loss [2], mainly because ice crystal formation and water migration that disrupt the gluten network [3]. Lipids, owing to their amphiphilic nature, play a crucial role in bakery formulations by retarding starch retrogradation and improving product quality. It was hypothesized that lipids with different saturation levels could modulate water mobility and gluten interactions, thereby influencing the structural integrity and quality stability of frozen dough. Therefore, this study compared representative liquid oils and solid fats to assess their effects on frozen dough texture, rheology, water distribution, thermal and structural properties, and bread quality, and to clarify how different lipids help maintain dough quality during freezing.

Lipids, mainly composed of triglycerides, are key ingredients in baking formulations that strongly affect dough rheology and bread quality. Supplementing with highly unsaturated oils—such as soybean (4%), corn (4%), peanut (3%), rapeseed (0–20%, *w*/*w* based on flour weight), and hempseed oils (0–15%)—has been shown to improve bread softness [4], reduce dough stickiness [5], and enhance extensibility. Moreover, these oils can modify the dough microstructure, enhancing gas retention [6] and improving bread volume, texture, and sensory quality. Solid fats also play a vital role in baking. Solid fats (270 g) influence the texture and rheology of dough and biscuits [7], and proper supplementation enhances baking performance. Adding shortening (35 g), coconut oil, or lard (2%) reduces dough hardness, improves elasticity, extends shelf life, and lowers water absorption [8], thereby improving overall baking performance [9]. Margarine or shortening (6 g) improves dough viscoelasticity and network stability through lipid–complex interactions [10]. Butter (30 g) reduces baking loss and crumb hardness while improving crust color [11], further optimizing the baking characteristics. These findings highlight the crucial role of both liquid oils and solid fats in modulating dough rheology and improving the quality of baked products.

Currently, most studies on lipid application in dough have focused on non-frozen systems, leaving their roles in frozen dough largely unexplored. Thus, lipid impacts on frozen dough and bread quality remain unclear. This study systematically compared the impacts of representative liquid oils (corn, peanut, soybean, and rapeseed) and solid fats (butter, shortening, margarine, lard, and coconut oil) on the quality of bread produced from frozen dough and its physicochemical characteristics. The findings provide insights into how various lipid types influence the characteristics of frozen dough and improve the final product quality. Moreover, these findings have practical significance, as selecting appropriate lipids can improve dough handling, reduce processing losses, extend shelf life, and support large-scale production of high-quality ready-to-bake bread.

## 2. Materials and Methods

### 2.1. Materials

Wheat flour was supplied by Henan Zhongrun Grain and Food Research Institute. Soybean oil, rapeseed oil, peanut oil, and shortening as the lipid sources (Yihai Kerry Foodsttuffs Marketing Co., Ltd., Wuhan, China); corn oil (Sanxing Corn Industry Technology Co., Ltd., Binzhou, China); butter (Fonterra Commercial Trading Co., Ltd., Wuhan, China); margarine (Namchow Food Co., Ltd., Tianjin, China); lard (Shuanghui Investment & Development Co., Ltd., Zhengzhou, China); and coconut oil (Hongshanhe Trading Co., Ltd., Haikou, China). Sucrose, salt, and yeast were obtained from suppliers in Wuhan. All chemicals were of analytical purity.

In the following sections, “F0” and “F2” represent samples that underwent zero and two cycles of freezing and thawing. The abbreviations “CK, CO, PO, SO, RO, BU, SH, MA, LA, and CC” correspond to the blank control, corn oil, peanut oil, soybean oil, rapeseed oil, butter, shortening, margarine, lard, and coconut oil, respectively.

### 2.2. Iodine Value (IV)

A precise amount of lipid, calculated based on the target iodine value, was weighed and fully mixed into a solution1:1 (*v*/*v*) of cyclohexane and glacial acetic acid, totaling 20 mL. Subsequently, Wijs reagent (25 mL) was supplemented, the solution was kept in a dark environment to ensure complete iodination. After incubation, potassium iodide solution (20 mL) and distilled water (150 mL) were supplemented. The released iodine was titrated using 0.1 M sodium thiosulfate until the yellow color nearly disappeared. Subsequently, three drops of starch indicator were introduced, and titration proceeded until the blue coloration vanished.(1)$$\mathrm{IV}=\frac{12.69\;\times \;c\left(V_{1}-V_{2}\right)}{m}$$

In the equation, c denotes the molar concentration of the sodium thiosulfate standard solution (mol/L), V_1_ and V_2_ denote the volumes consumed by the blank and sample (mL), and m represents the sample mass (g). All measurements were performed in triplicate, and the iodine value was calculated from the thiosulfate volume and reported as mean ± standard deviation (SD).

### 2.3. Preparation of the Frozen Dough

Dough was prepared according to the procedure described by Tingshi He [12] with minor modifications. The basic formulation of frozen dough consisted of wheat flour (100 g), salt (1 g), sugar (10 g), yeast (1.6 g), water (60 g), and lipid (3%, based on flour weight). Flour, sugar, salt, and water were initially combined and mixed at 100 rpm for 3 min to form dough, followed by high-speed mixing (200 rpm, 4 min). Lipid was then supplemented, and the dough was kneaded at 100 rpm for 2 min and then at 200 rpm for 11 min until smooth, uniform consistency was achieved. The dough was taken out, and left to rest before degassing, shaping, and dividing. Samples were stored at −18 °C for 23 h. Thawing was carried out in fermentation chamber (SP-16S, Sanyi Food Machinery Co., Ltd., Yancheng, China) at 33 °C and 75% relative humidity for 1 h, representing one freeze–thaw cycle. Samples were subjected to either 0 or 2 freeze–thaw cycles.

### 2.4. Texture Properties (TPA)

Dough texture was measured using a physical property analyzer (Food Technology Corporation, Sterling, VA, USA). Parameters measured included hardness, elasticity and chewiness. A P/36R cylindrical probe was used, trigger force of 5 g, and pre-test, test, and post-test speeds adjusted to 1 mm/s. Each 50 g sample was compressed to half of its original height.

### 2.5. Dynamic Rheological Rroperties

Rheological properties of frozen dough with lipids were measured using a multifunctional rheometer (Kinexus pro+, Malvern Instruments Ltd., Worcestershire, UK).

The parallel plate geometry with a 40 mm diameter was used, dough (5 g) was placed on the rheometer sample stage. The distance between the two plates was set to 2 mm. Tests were performed at 25 °C with a 0.5% strain (within the linear viscoelastic region, LVR). A frequency sweep from 0.1 to 10 Hz was conducted to evaluate the storage modulus (G′) and loss modulus (G″).

### 2.6. Water Distribution

Dough samples (5 g) were taken from the center and wrapped in plastic film to prevent moisture loss. Samples were placed in NMR tubes for low-field nuclear magnetic resonance (low-field nuclear magnetic resonance, Niumag, Shanghai, China) analysis. The parameters for measuring water distribution were as follows: spectral width (SW) 200 kHz, main frequency 20 MHz, repetition time (TW) 5000 ms, 90° pulse width (P1) 7.52 μs, 180° pulse width (P2) 14 μs, delay between 90° pulse and acquisition (RFD) 0.002 ms, receiver gain (RG1) 20, digital receiver gain (DRG1) 3, number of scans (NS) 8, echo time (TE) 0.2 ms, and number of echoes (NECH) 8000.

### 2.7. Differential Scanning Calorimetry

Differential scanning calorimetry (DSC) (TA Instruments Ltd., New Castle, DE, USA) analysis was performed to evaluate the thermal properties of freeze-dried dough. Briefly, Samples were mixed with deionized water (1:3, *w*/*w*) and equilibrated at 4 °C overnight. The mixture was heated from 0 °C to 100 °C at 10 °C/min using an empty pan as reference. The onset temperature (To), peak temperature (Tp), conclusion temperature (Tc) and enthalpy (ΔH) were recorded.

### 2.8. Fourier Transform Infrared Spectrometry (FT-IR)

The samples obtained in Section 2.3, first freeze-dried, then ground. The secondary structure of the frozen dough was analyzed using FTIR (Frontier, Perkin Elmer Co., High Wycombe, UK). Briefly, 2 mg of freeze-dried powder was mixed with 200 mg of KBr, and the mixture was finely ground and pressed into a transparent pellet. Spectral scans were performed over the range of 400–4000 cm^−1^, and the data analysis was performed using PeakFit_V4.12 software.

### 2.9. SEM

Freeze-dried dough slices were placed on the sample stage and sputter-coated with gold. The samples were then observed and imaged under SEM (SU8600, Hitachi, Ltd., Tokyo, Japan) with an applied voltage of 15 kV and a magnification set at 2000×.

### 2.10. Pasting Properties

#### 2.10.1. Preparation of Bread from Frozen Dough

Frozen dough was produced according to the formulation described in Section 2.3. After thawing, dough samples were proofed for 60 min in a proofing chamber (SP-16S, Sany Food Machinery Co., Ltd., Yancheng, China) under conditions of 35 °C and 80% relative humidity. The proofed dough was then baked in an oven with top and bottom temperatures of 200 °C and 180 °C, respectively, for 13 min, followed by a 2 h cooling period at room temperature before subsequent analyses.

#### 2.10.2. Textural Properties

Bread texture was measured following the same procedure as for dough (Section 2.4). Samples (20 mm × 20 mm × 30 mm) were collected from the central portion of the bread for analysis.

#### 2.10.3. Specific Volume (SV)

SV of bread prepared from frozen dough was determined according to Luo et al. [13]. The volume and weight of each bread sample were measured using a food volume analyzer (BVM-L370, Perten instruments, Hagersten, Sweden). SV was then calculated as the ratio of bread volume to weight.

#### 2.10.4. Sensory Evaluation

The sensory evaluation was conducted by nine trained members (four males and five females, aged 20–30) from the College of Food Science and Engineering. They assessed the bread samples for appearance, color, aroma, texture, and internal structure. All participants were informed of the study details, voluntarily participated, and consented to data collection. Their rights and privacy were protected. Ethical approval was not required, as no related risks were involved. The sensory evaluation form was designed according to the Chinese standard GB/T 20981—2021 [14] (Table 1).

### 2.11. Statistical Analysis

Each treatment was conducted in triplicate. Data were analyzed using SPSS 26.0 (IBM Inc., Chicago, IL, USA) analysis software using one-way analysis of variance (ANOVA) and Duncan’s multiple range test (*p* < 0.05). Results are expressed as mean ± standard deviation (mean ± SD).

## 3. Results and Discussion

### 3.1. IV of Different Lipid Types

Figure 1 shows the iodine values (IV) of the lipids. IV indicates the degree of unsaturation [15], higher IV means greater unsaturation. Liquid oils (8.76–137.81 g/100 g) showed higher IVs than solid fats, in the orde: SO > CO > RO > PO > LA > MA > SH > BU > CC. Significant variations in IV were observed, likely due to differences in raw materials and lipid freshness. Among solid fats, LA had the highest IV (71.46 g/100 g), reflecting a higher level of unsaturation. Lipid unsaturation strongly influences dough structure and bread quality. Unsaturated oils improve gluten interaction and gas retention, while solid fats limit water–protein binding and alter ice crystallization. The iodine value thus indicates both lipid composition and its relationship to bread quality.

### 3.2. Textural Properties of Frozen Dough

The textural properties of the dough are shown in Table 2. At F0, lipid supplementation reduced dough hardness due to hydrophobic interactions between lipids and gluten. Liquid oils rich in unsaturated fatty acids (e.g., CO, IV = 121.92 g/100 g) penetrated gluten’s hydrophobic regions, limiting protein aggregation and softening the dough [16]. In contrast, solid fats (BU and SH), rich in saturated fatty acids, formed weaker lipid–gluten complexes and showed slightly higher hardness, reflecting the lower flexibility of saturated fatty acid chains. After F2, dough hardness increased for all samples, with variations depending on lipid type. Liquid-oil group, particularly those rich in unsaturated fatty acids (e.g., CO), showed larger increases (from 10.68 to 15.20%) in hardness due to their lower water-binding capacity and looser gluten–lipid interfaces, which promote large ice crystal formation and gluten damage. In contrast, solid fats such as SH and CC limited water migration through their crystalline structures; the β′-crystal networks in CC further acted as barriers to protect gluten integrity.

The results of dough elasticity and chewiness showed that all lipid-supplemented doughs exhibited lower elasticity than F0-CK, confirming that freeze–thaw cycles impaired gluten extensibility [2]. Among liquid oils, PO exhibited the highest springiness (13.12 mm) at F0 due to its moderate unsaturation (IV = 90.45 g/100 g), which promoted balanced lipid–gluten interactions. After F2, its elasticity slightly decreased, whereas highly unsaturated CO showed a larger reduction (from 11.74 to 10.77 mm) because its loose gluten network was more vulnerable to ice crystal damage [17]. In solid-fat group, SH maintained stable springiness (from 12.29 to 12.47 mm) because of its rigid hydrogenated matrix, while CC showed increased springiness (from 12.24 to 16.79 mm), likely resulting from partial recrystallization and water reabsorption during thawing that temporarily plasticized the gluten network. Chewiness increased in all F2 samples and correlated positively with hardness. However, doughs containing solid fats (BU: from 97.87 to 108.78 mj) exhibited lower chewiness than CK (from 132.65 to 163.48 mj), as their fine, evenly dispersed fat crystals acted as fillers that mitigated gluten brittleness [18]. Overall, results of hardness, elasticity, and chewiness demonstrated that solid fats conferred greater freeze–thaw resistance to dough than liquid oils.

### 3.3. Dynamic Rheological Properties

Rheological properties of dough are shown in Figure 2. Figure 2A,B show impacts of liquid oils (CO, PO, SO, and RO) on the viscoelastic behavior of frozen dough. At F0, both G′ and G″ decreased compared with F0-CK, as the flexible unsaturated chains of liquid oils disrupted gluten cross-linking and reduced network rigidity. RO showed the lowest values, consistent with Chen et al. [19]. After F2, all oil-treated doughs exhibited further decreases in G′ and G″ due to freeze–thaw-induced water migration and ice crystal damage [20]. Compared with F2-CK, CO, PO, and RO showed greater viscoelastic losses, while SO exhibited less reduction, likely due to its higher vitamin E and β-sitosterol contents that mitigate gluten oxidation and enhance freeze–thaw stability.

Figure 2C,D show that doughs containing solid fats exhibited rheological trends similar to those of liquid-oil group but maintained better structural stability after freeze–thaw cycles. At F0, SH was closest to F0-CK, with relatively high modulus values, which can be attributed to differences in fatty acid composition and viscosity. SH and BU formed stable crystalline networks that reinforced the gluten structure, resulting in modulus values comparable to CK, whereas low-melting-point fats (such as CC) exhibited weaker crystallinity and the lowest modulus values. After F2, both G′ and G″ decreased in solid-fat group, indicating that freeze–thaw damage still occurred but to a lesser extent than in the liquid oil and control groups. This improvement is mainly due to the formation of uniform β′ crystals by hydrogenated fats (BU, SH, MA) [21], which limit water migration and inhibit large ice crystal growth, thus protecting the gluten network. In contrast, the CC dough showed an opposite trend, with marked increases in G′ and G″ after F2, likely due to its higher unsaturated fatty acid content leading to incomplete crystallization. Overall, solid fats exhibited superior freeze–thaw stability by stabilizing crystalline structures and reducing water migration.

### 3.4. Moisture Migration of Frozen Dough

The results of moisture distribution of frozen dough are shown in Table 3. Water distribution strongly influences dough processing and product quality. LF-NMR was used to evaluate water distribution in frozen dough with different lipids. Three relaxation peaks were detected: bound water (T21), immobilized water (T22), and free water (T23) [22]. A21, A22, A23 represent the relative proportions of these water states [23]. As shown in Table 3, weakly bound water dominated in all samples. At F0, lipid supplementation increased A23, likely due to hydrophobic interactions with gluten and starch that converted some bound or immobilized water into free water. After F2, CK showed increased A21 (from 20.82% to 21.24%), slight decrease in A22 (from 78.75% to 78.50%), and reduced in A23 (from 0.44% to 0.27%), indicating ice crystal formation damaged the gluten network, while thawing redistributed water, increasing bound water and decreasing free water.

F2 in the liquid-oil group reduced A21 and increased A22 and A23, indicating that part of the strongly bound water was converted into weakly bound water. This likely results from a hydrophobic film at the starch–protein interface that limits water binding. RO showed the smallest changes (A22: from 77.47 to 77.80%; A23: from 1.81 to 1.88%), demonstrating its superior ability to restrict water migration and protect the gluten network. In solid-fat group, F2 treatment resulted in lower A21 but higher A22 and A23 values. Specifically, A22 decreased slightly in CK (from 78.75 to 78.50%), but increased in the BU-supplemented dough (from 77.37 to 79.09%), accompanied by a rise in A23 (from 1.19 to 1.34%). SH-supplemented dough showed increases in A22 (from 77.50 to 78.72%, close to F2-CK) and A23 (from 1.09 to 1.30%), reflecting the β′-crystal network that limits water migration [24]. Overall, liquid oils promoted the formation of free water and enhanced dough extensibility, whereas solid fats, with their stable three-dimensional crystal structures, restricted water migration and ice crystal formation, thus preserving dough processability and product quality.

### 3.5. DSC

Thermal properties of the dough are shown in Table 4. To, Tp, and Tc reflect starch stability [25]. Compared with the CK, lipid supplementation reduced the ΔH [26]. The decrease in ΔH was generally greater in liquid-oil group than solid-fat group, indicating that liquid oils offered weaker protection to the gluten–starch network during freezing. Among them, PO and RO had the lowest ΔH values after F2 (3.40 and 3.41 J/g, respectively), likely due to their high fatty acid unsaturation causing protein unfolding and reduced hydrogen bonding. In contrast, CO showed less thermal degradation, possibly owing to its moderate crystallinity and emulsifying properties. In solid-fat group, ΔH remained relatively stable—especially for SH—indicating its ability to form a stable β′-crystalline network and thus reinforce the gluten–starch matrix. Overall, DSC results show that solid fats outperform liquid oils in maintaining the thermal stability of frozen dough, which aligns with the FTIR findings that solid-fat systems exhibited less disruption of protein secondary structure.

### 3.6. FT-IR

FTIR spectroscopy was used to analyze protein secondary structure, focusing on the amide I band (1600–1700 cm^−1^) [27], mainly arising from C=O stretching of peptide bonds [28] (Figure 3). The amide I band was deconvoluted using Peakfit 4.12 (second-derivative fitting), and peak areas were integrated to calculate the percentage of each secondary structure. Table 5 shows the variations in β-sheet and β-turn contents in doughs after F0 and F2 treatments.

In liquid-oil group, F2 increased β-sheet content (Lipid addition may change gluten-protein aggregation, leading to a more ordered secondary structure at the oil/water interface [29]) and decreased β-turns, with variations among oils. In the CO dough, the β-sheet content increased from 24.69 to 25.95%, while the β-turn content decreased from 29.80 to 24.71%. SO showed more pronounced structural changes: β-sheet content increased from 24.13 to 25.88%, whereas β-turn content decreased from 28.86 to 25.77%. In contrast, RO showed a smaller increase in β-sheet structures compared with SO, likely due to oxidation of its polyunsaturated fatty acids during freeze–thaw cycles, which weakens lipid–protein interactions and protein structure stability.

In solid-fat group, BU showed nearly unchanged secondary structures after freeze–thaw cycles, with β-sheet (from23.46 to 23.94%) and β-turn (from 26.44 to 27.45%) contents remaining nearly constant, likely due to its stable β′-crystalline network that preserved the native gluten conformation. In contrast, the highly unsaturated LA (IV = 71.46 g/100 g) showed a marked decrease in β-sheets and an increase in β-turns (from 27.86 to 29.51%), as poor crystallization efficiency disrupted the β-sheet (from 26.43 to 24.83%) structures and promoted its conversion to β-turns, resembling the pattern observed in liquid oils.

### 3.7. Microstructure of Frozen Dough

Dough comprises a starch–protein network, and its microstructural changes after freeze–thaw cycles are shown in Figure 4. In F0-CK, starch granules were uniformly embedded in a compact gluten matrix with fine, evenly distributed pores. After F2, large ice crystals formed from free water disrupted the gluten network, and subsequent melting caused dehydration and weakened cross-linking [30], leading to a loose and disordered structure. After F2, liquid oil-supplemented doughs (CO, PO, SO, RO) showed greater microstructural damage, especially SO, where the gluten network was fragmented and large pores formed. This is because liquid oils (rich in unsaturated fatty acids) form only loose hydrophobic interactions with gluten proteins. Their flexible hydrocarbon chains are inserted into gluten hydrophobic domains but fail to form stable cross-links, resulting in a weak gluten-lipid composite structure [31]. During freeze–thaw, the hydrophobic film formed by liquid oils is easily disrupted by ice crystal growth—this not only fails to inhibit water migration but also reduces the mechanical strength of the gluten network, exacerbating structural collapse. In contrast, RO-supplemented dough retained relatively better structural integrity after F2. Solid fat-supplemented doughs (BU, SH, MA, LA, CC) exhibited significantly better microstructural stability after freeze–thaw cycles than liquid-oil group (Figure 4B). Among them, SH-supplemented dough showed the strongest freeze–thaw resistance, due to its stable three-dimensional β′ crystalline network [32] with a compact structure that minimizes moisture loss, SH forms stronger hydrophobic interactions between the saturated fatty acids in solid fats and gluten proteins.

### 3.8. Bread Baking Properties

#### 3.8.1. Textural Properties

TPA is an important indicator of bread quality [33], and softness is a key factor influencing consumer preference. The textural properties of bread are shown in Figure 5. The CK group showed the largest decrease in hardness (from 2.65 N to 1.49 N), indicating severe gluten network disruption after freeze–thaw cycles. In contrast, breads with liquid oils (CO, PO, SO, RO) showed better structural stability. CO showed the smallest decline (from 0.74 to 0.68 N) due to its high unsaturation (IV = 121.92 g/100 g), which promotes uniform starch–lipid complex formation and inhibits starch retrogradation. PO showed the largest reduction (from 1.10 to 0.82 N), consistent with its highest IV among the liquid oils. Among the solid fats, SH produced the softest bread and exhibited the highest freeze–thaw stability, with only an 11.5% decrease in hardness after two cycles (from 1.04 N to 0.92 N), despite its overall performance being slightly lower than that of the liquid oils. SH, a hydrogenated solid fat, forms a compact crystalline network that minimizes ice crystal damage to the gluten matrix, preserves the porous crumb structure, and reduces hardness. In contrast, LA (highest IV among solid fats) showed the largest decrease in hardness (from 1.42 to 0.94 N), owing to poor crystallization efficiency and increased moisture migration. Overall, CO and SH effectively preserved bread softness—CO through enhanced lipid–starch interactions and SH through a stable crystalline network—together improving the quality of frozen-dough bread.

#### 3.8.2. Specific Volume of Frozen Bread

Figure 6 shows SV of breads prepared from frozen doughs with different lipids. Unlike previous findings that reported SV reduction after freeze–thaw treatment [34]. lipid supplementation in this study increased SV, likely due to the lubricating effect of lipids that reduced friction between gluten and starch components [32].

In liquid-oil group, the SV of bread at F0 was significantly higher than that of F0-CK, following the order: F0-RO > F0-CO > F0-PO > F0-SO > F0-CK. This may result from a hydrophobic film formed by liquid oils at the gluten–starch interface, which maintains yeast activity and improves gas retention. RO consistently showed the highest SV, and a 3% increase after F2 (from 5.42 to 5.57 mL/g) indicates that its high oleic acid content [35], may protect the gluten network from oxidative damage. In solid-fat group, the SV at F0 followed the order F0-SH > F0-BU > F0-MA > F0-LA > F0-CC > F0-CK, with all lipid-containing samples showing higher values than CK. After F2, the SV decreased below that of CK, following the order F2-CK > F2-BU > F2-SH > F2-MA > F2-LA > F2-CC. The lowest SV observed for LA and CC may result from large crystalline structures disrupting the gluten network, whereas BU and SH maintained higher SV due to their smaller and more stable crystals, which exhibited greater resistance to freeze–thaw recrystallization and minimized gluten damage.

#### 3.8.3. Sensory Evaluation of Bread

Sensory evaluation results are shown in Figure 7, where appearance, color, odor, texture and structure were each scored out of 20 points. F2-CK obtained the lowest score, indicating that freeze–thaw cycles impaired the sensory quality of bread [36]. After F2, among liquid oils, CO and PO achieved the highest sensory scores; among solid fats, SH exhibited superior sensorzy performance, with the highest scores in appearance and texture at F0. This phenomenon is mainly attributed to the β′-crystal network of SH, which can effectively inhibit ice crystal growth and stabilize the gluten structure. Even after F2, they still maintain softness and dense structure of bread. Pearson correlation analysis was performed to examine the relationships among textural properties, SV, and sensory evaluation (Figure 8). Sensory scores were significantly negatively correlated with the hardness and chewiness of bread (r = −0.47 and −0.46) (*p* < = 0.05), indicating that higher hardness and chewiness led to lower sensory acceptability.

## 4. Conclusions

This study investigated the impacts of four liquid oils (CO, PO, SO, RO) and five solid fats (BU, SH, MA, LA, CC) on frozen dough and bread quality. Doughs with solid fats retained higher G′ and G″ after freeze–thaw, showing better viscoelasticity, while moisture analysis indicated improved freeze–thaw stability and reduced ice crystal damage. FTIR showed lipid addition preserved β-sheet gluten structures, and texture analysis highlighted CO and SH in maintaining bread softness. DSC revealed solid fats better maintained dough thermal stability. These findings guide lipid selection in frozen bakery formulations to enhance stability and shelf life. Further studies should explore varied lipid levels, additional freeze–thaw cycles, and industrial-scale conditions.

## Figures and Tables

**Figure 1 foods-14-04032-f001:**
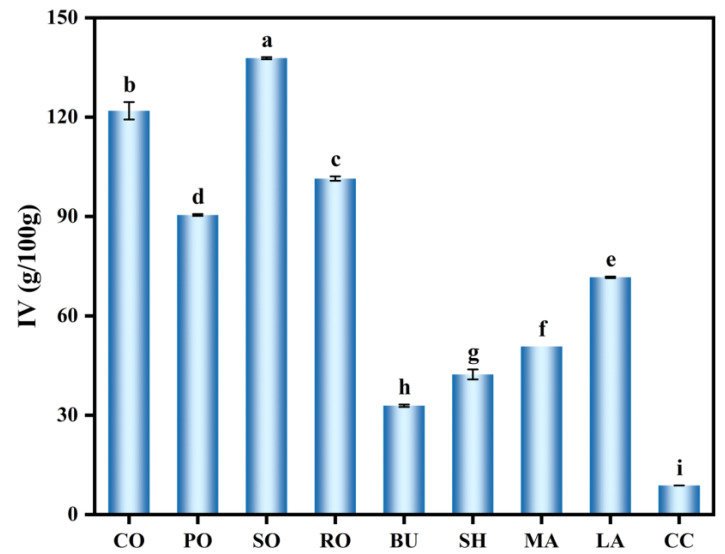
IV of different lipid types. Different letters within a column denote significant differences (*p* < 0.05).

**Figure 2 foods-14-04032-f002:**
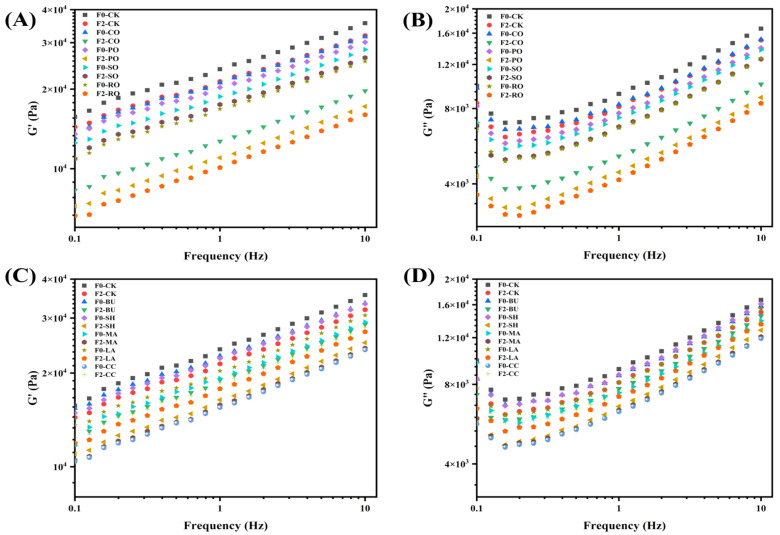
Impact of different lipid types on rheological properties of frozen dough. Liquid oils, (**A**,**B**) represent G′ and G″; solid fats, (**C**,**D**) represent G′ and G″.

**Figure 3 foods-14-04032-f003:**
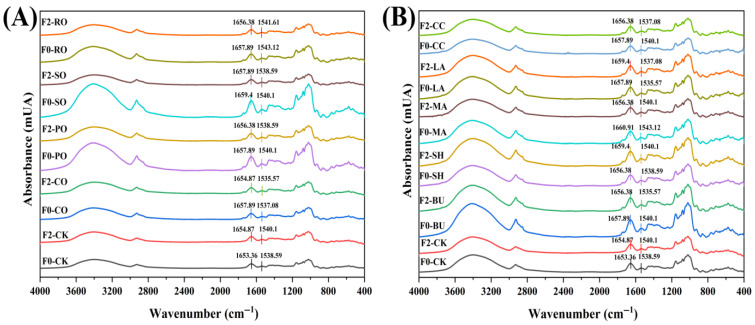
FTIR spectra of frozen doughs with different lipid supplementation. (**A**) Liquid oil group; (**B**) Solid fat group.

**Figure 4 foods-14-04032-f004:**
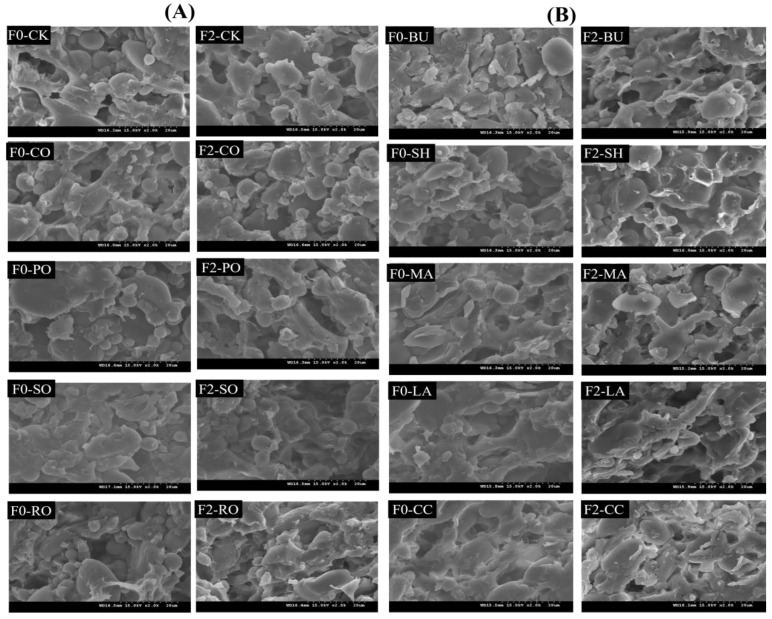
Impact of different lipid types on the microstructure of frozen dough. (**A**) Liquid oil group; (**B**) Solid fat group.

**Figure 5 foods-14-04032-f005:**
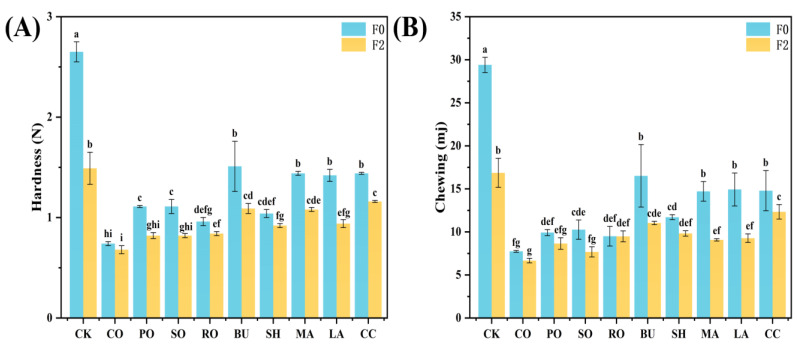
Impact of different lipid types on textural properties of bread. (**A**) Hardness; (**B**) Chewiness. Different letters within a column denote significant differences (*p* < 0.05).

**Figure 6 foods-14-04032-f006:**
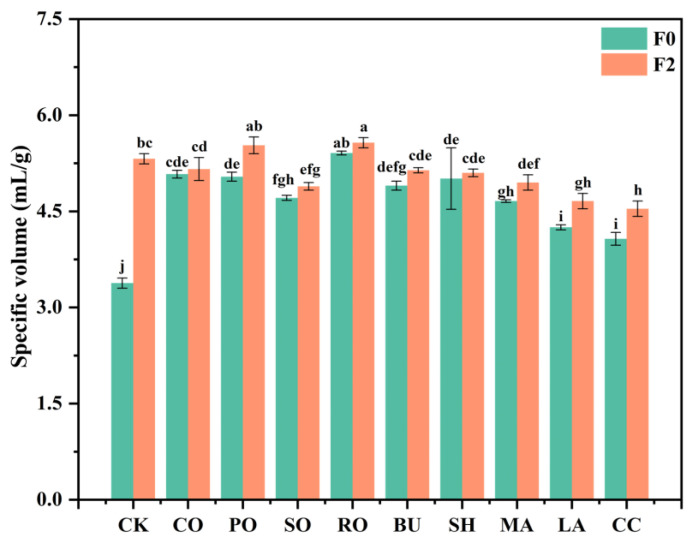
Impact of different lipid types on specific volume of bread. Results are shown as mean ± SD. Different letters within a column denote significant differences (*p* < 0.05).

**Figure 7 foods-14-04032-f007:**
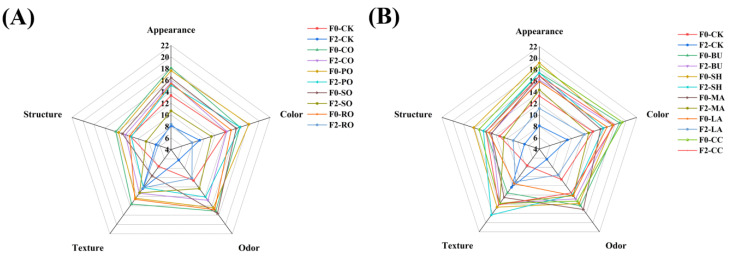
Sensory analysis of frozen bread. (**A**) Liquid oil group; (**B**) Solid fat group.

**Figure 8 foods-14-04032-f008:**
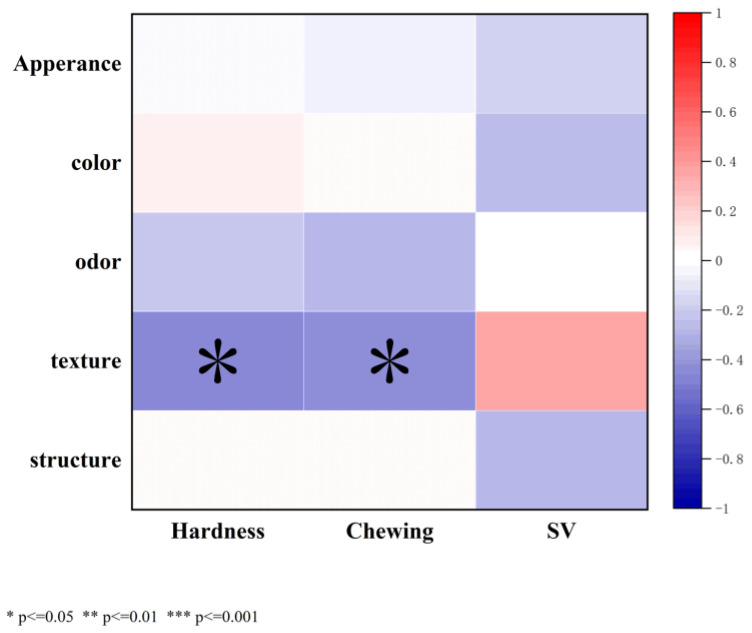
Correlation heatmap of bread quality parameters.

**Table 1 foods-14-04032-t001:** Sensory evaluation of frozen bread.

Sensory Characteristics	Score	Definition
Appearance	20	Smooth surface without cracks. 16–20 points
		Smooth surface with minor cracks. 11–15 points
		Collapsed with visible cracks. 1–10 points
Color	20	The crust has a uniform color and good sheen. 16–20 points
		The crust is light golden with a moderate sheen. 11–15 points
		The crust is burnt. 1–10 points
Odor	20	A rich and pleasant oily fragrance. 16–20 points
		A slightly faint oily fragrance. 11–15 points
		Rancid and putrid oily odors. 1–10 points
Texture	20	Soft, non-sticky, smooth. 16–20 points
		Slightly sticky/firm. 11–15 points
		Too hard/sticky, hard to swallow. 1–10 points
Structure	20	Even pores, thin intact walls. 16–20 points
		Few large pores, medium walls. 11–15 points
		Oversized pores, broken loose walls. 1–10 points

**Table 2 foods-14-04032-t002:** Impact of different lipid types on texture properties of frozen dough (*n* = 3).

Lipid Types	Number of Cycles		Hardness (N)	Elasticity (mm)	Chewability (mj)
—	F0	CK	12.23 ± 0.30 ^d^	14.32 ± 0.67 ^b^	132.65 ± 16.71 ^cde^
	F2	CK	15.40 ± 0.71 ^b^	12.83 ± 0.67 ^cd^	163.48 ± 4.99 ^a^
Liquid oils	F0	CO	10.68 ± 0.14 ^g^	11.74 ± 0.27 ^ef^	87.90 ± 3.67 ^ij^
	F2	CO	15.20 ± 0.24 ^b^	10.77 ± 1.14 ^g^	118.82 ± 13.87 ^defgh^
	F0	PO	10.94 ± 0.06 ^fg^	13.12 ± 0.40 ^c^	99.34 ± 8.62 ^hij^
	F2	PO	14.68 ± 0.56 ^c^	12.65 ± 0.61 ^cde^	133.93 ± 9.50 ^cd^
	F0	SO	11.55 ± 0.33 ^e^	11.06 ± 0.79 ^fg^	82.68 ± 5.03 ^j^
	F2	SO	15.66 ± 0.21 ^b^	12.96 ± 0.17 ^cd^	157.25 ± 13.57 ^ab^
	F0	RO	11.33 ± 0.46 ^ef^	12.15 ± 0.00 ^cde^	111.11 ± 18.94 ^fghi^
	F2	RO	14.30 ± 0.41 ^c^	12.46 ± 1.00 ^cde^	123.26 ± 17.82 ^cdefg^
Solid fats	F0	BU	11.46 ± 0.36 ^ef^	12.26 ± 0.72 ^cde^	97.87 ± 10.22 ^hij^
	F2	BU	14.30 ± 0.25 ^c^	12.46 ± 0.98 ^cde^	108.78 ± 14.39 ^efgh^
	F0	SH	12.17 ± 0.13 ^d^	12.29 ± 0.30 ^cde^	106.72 ± 3.57 ^ghi^
	F2	SH	14.35 ± 0.29 ^c^	12.47 ± 0.24 ^cde^	140.97 ± 24.96 ^bcd^
	F0	MA	11.72 ± 0.15 ^de^	12.06 ± 0.21 ^de^	102.71 ± 4.02 ^ghij^
	F2	MA	14.61 ± 0.22 ^c^	12.84 ± 0.18 ^cd^	131.01 ± 18.95 ^cdef^
	F0	LA	11.48 ± 0.06 ^ef^	11.94 ± 0.07 ^def^	119.01 ± 0.77 ^defgh^
	F2	LA	16.41 ± 0.12 ^a^	11.97 ± 0.28 ^def^	141.87 ± 5.27 ^bc^
	F0	CC	11.84 ± 0.32 ^de^	12.24 ± 0.11 ^cde^	103.11 ± 10.68 ^ghij^
	F2	CC	14.20 ± 0.22 ^c^	16.79 ± 0.24 ^a^	142.82 ± 13.89 ^abc^

Note: “—” indicates no lipid supplementation. Results are shown as mean ± SD. Different letters within a column denote significant differences (*p* < 0.05).

**Table 3 foods-14-04032-t003:** Impact of different lipid types on water distribution of frozen dough (*n* = 3).

Lipid Types	Number of Cycles		T21 (ms)	T22 (ms)	T23 (ms)	A21 (%)	A22 (%)	A23 (%)
—	F0	CK	1.32 ± 0.00	16.30 ± 0.00	231.01 ± 0.00	20.82 ± 0.12 ^bcde^	78.75 ± 0.11 ^ab^	0.44 ± 0.03 ^k^
	F2	CK	1.15 ± 0.000	16.30 ± 0.00	265.61 ± 0.00	21.24 ± 0.65 ^ab^	78.50 ± 0.65 ^bc^	0.27 ± 0.01 ^l^
Liquid oils	F0	CO	1.32 ± 0.00	16.30 ± 0.00	200.92 ± 0.00	21.22 ± 0.25 ^ab^	77.00 ± 0.20 ^k^	1.77 ± 0.05 ^d^
	F2	CO	1.32 ± 0.00	16.30 ± 0.00	200.92 ± 0.00	20.47 ± 0.23 ^defg^	77.70 ± 0.24 ^efghi^	1.84 ± 0.02 ^cd^
	F0	PO	1.32 ± 0.00	16.30 ± 0.00	200.92 ± 0.00	20.91 ± 0.07 ^bcd^	77.25 ± 0.08 ^jk^	1.84 ± 0.10 ^cd^
	F2	PO	1.32 ± 0.00	16.30 ± 0.00	174.75 ± 0.00	20.44 ± 0.11 ^efg^	77.69 ± 0.10 ^efghi^	1.87 ± 0.05 ^cd^
	F0	SO	1.32 ± 0.00	16.30 ± 0.00	200.92 ± 0.00	20.72 ± 0.20 ^cdef^	77.29 ± 0.19 ^ijk^	1.99 ± 0.02 ^b^
	F2	SO	1.32 ± 0.00	16.30 ± 0.00	200.92 ± 0.00	20.13 ± 0.09 ^gh^	77.78 ± 0.04 ^efgh^	2.08 ± 0.06 ^a^
	F0	RO	1.32 ± 0.00	16.30 ± 0.00	200.92 ± 0.00	20.72 ± 0.23 ^cdef^	77.47 ± 0.25 ^ghij^	1.81 ± 0.06 ^cd^
	F2	RO	1.32 ± 0.00	16.30 ± 0.00	174.75 ± 0.00	20.31 ± 0.07 ^fgh^	77.80 ± 0.04 ^efg^	1.88 ± 0.09 ^c^
Solid fats	F0	BU	1.32 ± 0.00	16.30 ± 0.00	200.92 ± 0.00	21.45 ± 0.06 ^a^	77.37 ± 0.06 ^hijk^	1.19 ± 0.05 ^h^
	F2	BU	1.32 ± 0.00	16.30 ± 0.00	174.75 ± 0.00	19.54 ± 0.10 ^j^	79.09 ± 0.10 ^a^	1.34 ± 0.03 ^f^
	F0	SH	1.32 ± 0.00	16.30 ± 0.00	200.92 ± 0.00	21.41 ± 0.07 ^a^	77.50 ± 0.09 ^ghij^	1.09 ± 0.06 ^i^
	F2	SH	1.32 ± 0.00	16.30 ± 0.00	174.75 ± 0.00	19.99 ± 0.54 ^hi^	78.72 ± 0.50 ^ab^	1.30 ± 0.05 ^fg^
	F0	MA	1.15 ± 0.00	16.30 ± 0.00	174.75 ± 0.00	20.57 ± 0.17 ^cdefg^	77.95 ± 0.08 ^def^	1.49 ± 0.09 ^e^
	F2	MA	1.15 ± 0.00	16.30 ± 0.00	174.75 ± 0.00	19.69 ± 0.20 ^ij^	78.67 ± 0.09 ^b^	1.53 ± 0.03 ^e^
	F0	LA	1.32 ± 0.00	16.30 ± 0.00	200.92 ± 0.00	21.40 ± 0.19 ^a^	77.66 ± 0.21 ^fghij^	0.94 ± 0.03 ^j^
	F2	LA	1.32 ± 0.00	16.30 ± 0.00	200.92 ± 0.00	20.55 ± 0.12 ^cdefg^	78.22 ± 0.11 ^cd^	1.23 ± 0.02 ^gh^
	F0	CC	1.32 ± 0.00	16.30 ± 0.00	200.92 ± 0.00	21.53 ± 0.18 ^a^	77.30 ± 0.14 ^ijk^	1.17 ± 0.04 ^hi^
	F2	CC	1.32 ± 0.00	16.30 ± 0.00	200.92 ± 0.00	20.96 ± 0.05 ^bc^	78.09 ± 0.06 ^de^	0.95 ± 0.02 ^j^

Note: “—” indicates no lipid supplementation. Results are shown as mean ± SD. Different letters within a column denote significant differences (*p* < 0.05).

**Table 4 foods-14-04032-t004:** Impact of different lipid types on thermal properties of frozen dough (*n* = 3).

Lipid Types	Number of Cycles		T_O_ (°C)	T_P_ (ms)	T_C_ (ms)	ΔH (J·g)
—	F0	CK	58.09 ± 0.51 ^f^	64.97 ± 0.21 ^cdefg^	71.21 ± 0.41 ^ab^	4.39 ± 0.28 ^a^
	F2	CK	59.86 ± 0.41 ^bcde^	65.08 ± 0.09 ^cdef^	70.85 ± 1.31 ^abc^	4.24 ± 0.36 ^ab^
Liquid oils	F0	CO	60.62 ± 0.83 ^abcd^	65.61 ± 0.18 ^abc^	71.54 ± 0.93 ^a^	3.54 ± 0.14 ^c^
	F2	CO	61.06 ± 0.79 ^ab^	65.29 ± 0.260 ^bcd^	70.99 ± 0.67 ^abc^	3.52 ± 0.33 ^cd^
	F0	PO	61.25 ± 0.95 ^a^	65.84 ± 0.56 ^ab^	70.53 ± 0.89 ^abc^	3.02 ± 0.17 ^de^
	F2	PO	60.96 ± 0.43 ^abc^	65.25 ± 0.06 ^bcde^	70.99 ± 0.14 ^abc^	3.40 ± 0.13 ^cd^
	F0	SO	59.55 ± 0.90 ^de^	64.79 ± 0.30 ^defg^	70.95 ± 0.68 ^abc^	3.81 ± 0.11 ^bc^
	F2	SO	60.06 ± 0.49 ^abcd^	64.60 ± 0.22 ^efg^	70.63 ± 0.15 ^abc^	3.48 ± 0.26 ^cd^
	F0	RO	59.72 ± 1.45 ^cde^	64.48 ± 0.35 ^fg^	69.91 ± 0.48 ^bc^	2.87 ± 0.21 ^e^
	F2	RO	61.17 ± 0.61 ^a^	65.27 ± 0.22 ^bcde^	71.46 ± 0.96 ^a^	3.41 ± 0.10 ^cd^
Solid fats	F0	BU	58.70 ± 0.35 ^ef^	64.38 ± 0.00 ^g^	69.65 ± 0.02 ^c^	3.44 ± 0.38 ^cd^
	F2	BU	59.56 ± 0.38 ^de^	65.22 ± 0.24 ^bcde^	70.85 ± 0.73 ^abc^	3.80 ± 0.06 ^bc^
	F0	SH	60.71 ± 0.50 ^abcd^	65.30 ± 0.31 ^bcd^	70.86 ± 0.86 ^abc^	3.58 ± 0.35 ^c^
	F2	SH	60.95 ± 0.89 ^abc^	66.13 ± 0.92 ^a^	71.70 ± 0.27 ^a^	3.58 ± 0.33 ^c^
	F0	MA	60.86 ± 0.67 ^abc^	65.50 ± 0.25 ^abc^	71.62 ± 0.65 ^a^	3.70 ± 0.54 ^c^
	F2	MA	60.63 ± 0.43 ^abcd^	65.56 ± 0.54 ^abc^	71.17 ± 0.71 ^ab^	3.80 ± 0.22 ^bc^
	F0	LA	60.45 ± 0.63 ^abcd^	65.19 ± 0.24 ^bcde^	71.44 ± 1.06 ^a^	3.71 ± 0.03 ^c^
	F2	LA	59.87 ± 0.21 ^bcde^	64.99 ± 0.20 ^cdefg^	71.24 ± 0.88 ^ab^	3.44 ± 0.20 ^cd^
	F0	CC	60.61 ± 0.25 ^abcd^	65.23 ± 0.23 ^bcde^	71.35 ± 0.88 ^ab^	3.50 ± 0.31 ^cd^
	F2	CC	60.11 ± 0.19 ^abcd^	65.09 ± 0.40 ^cdef^	70.48 ± 0.91 ^abc^	3.84 ± 0.22 ^bc^

Note: “—” indicates no lipid addition. Results are shown as mean ± SD. Different letters within a column denote significant differences (*p* < 0.05).

**Table 5 foods-14-04032-t005:** Impact of different lipid types on secondary structure of frozen dough (*n* = 3).

Lipid Types	Number of Cycles		β-Sheet (%)	β-Turn (%)
—	F0	CK	25.31 ± 0.54 ^abc^	25.61 ± 0.26 ^ef^
	F2	CK	24.80 ± 0.89 ^bcd^	26.88 ± 1.94 ^cdef^
Liquid oils	F0	CO	24.69 ± 0.13 ^bcd^	29.80 ± 0.20 ^a^
	F2	CO	25.95 ± 0.32 ^ab^	24.71 ± 1.67 ^f^
	F0	PO	24.80 ± 1.61 ^bcd^	26.20 ± 0.58 ^def^
	F2	PO	25.23 ± 0.22 ^abc^	29.26 ± 0.35 ^ab^
	F0	SO	24.13 ± 0.83 ^cd^	28.86 ± 1.93 ^abc^
	F2	SO	25.88 ± 0.20 ^ab^	25.77 ± 0.36 ^def^
	F0	RO	24.87 ± 0.17 ^bc^	29.77 ± 0.13 ^def^
	F2	RO	25.76 ± 0.67 ^ab^	29.11 ± 1.01 ^ab^
Solid fats	F0	BU	23.46 ± 0.67 ^d^	26.44 ± 0.34 ^def^
	F2	BU	23.94 ± 1.14 ^cd^	27.45 ± 2.21 ^bcde^
	F0	SH	25.75 ± 0.20 ^ab^	25.87 ± 0.34 ^def^
	F2	SH	25.58 ± 0.25 ^ab^	27.60 ± 1.30 ^bcde^
	F0	MA	25.70 ± 0.09 ^ab^	26.04 ± 0.12 ^d^
	F2	MA	25.80 ± 0.46 ^ab^	27.38 ± 2.66 ^bcde^
	F0	LA	26.43 ± 1.34 ^a^	27.86 ± 1.47 ^abcd^
	F2	LA	24.83 ± 0.11 ^bcd^	29.51 ± 0.22 ^ab^
	F0	CC	25.04 ± 0.58 ^abc^	26.54 ± 0.53 ^def^
	F2	CC	25.14 ± 0.77 ^abc^	25.89 ± 0.51 ^def^

Note: “—” indicates no lipid supplementation. Results are shown as mean ± SD. Different letters within a column denote significant differences (*p* < 0.05).

## Data Availability

The data presented in this study are available on request from the corresponding author.

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
