# Peer review of "Impact of Different Types of Lipids on the Quality of Frozen Dough and Bread"

_foods, 2025, doi:10.3390/foods14234032_

Round 1

Reviewer 1 Report

Comments and Suggestions for Authors

The manuscript presents a study on the effect of different types of lipids on the quality of frozen dough and bread. The observations are detailed below:
1. The title and abstract are adequate, but the abstract should include the experimental design and a final sentence about the study's innovative contribution to the food industry. In the case of keywords, avoid using the same ones as in the title.
2. The first paragraph of the introduction should better capture the reader's attention, include an explicit hypothesis that guides the experimental approach, and improve the presentation of the specific objectives (the use of bullet points is recommended).
3. The technical specifications of the equipment used should be included, as well as the purity or grade of the reagents used. 
4. The experimental design applied should be explicitly stated, as well as the statistical analyses used for that experimental design.
5. Tables and figures should include more quantitative comparative analyses, for example, establishing correlations between the iodine value (IV) and the texture parameters of bread or dough.
6. The number of freezing cycles evaluated should be increased, and a complete sensory evaluation should be incorporated to allow the instrumental results to be related to consumer perception.
7. Additional key instrumental measurements should be included, such as DSC, FTIR, XRD, colorimetry, Aw, and CLSM, which will strengthen the interpretation and correlation between structure, texture, and bread quality.
8. The discussions are mostly descriptive and require a better interpretation of the lipid-protein-water interaction mechanisms. Scientific depth should be strengthened through a more analytical discussion that relates the experimental results to specific physicochemical mechanisms (lipid-protein interactions, water migration, ice crystal formation, and changes in the gluten network). 
9. The contextualization of the findings in relation to previous studies (preferably from the last 5 years) needs to be improved.
10. The technological and industrial scalability implications should be discussed in greater detail.
11. Improve the overall presentation of the manuscript through more fluid organization and better integrated comparative figures (we recommend reviewing the FOODS author guidelines and using the Word template and Endnote plug-in for citations).
12. The wording of the conclusions should be improved to make them more interpretively robust. Also include the limitations of the study and possible future lines of research, as well as its specific application in the food industry.
13. Reduce the Ithenticate similarity index (20%).

Comments on the Quality of English Language

It has long sentences; review by a native speaker editor is recommended, standardize terms, and avoid repetitive connectors that reduce clarity.

Author Response

Please see the attachment for the detailed responses to all reviewer comments. We have carefully addressed each point and made corresponding revisions in the manuscript. All changes are highlighted in red. We sincerely appreciate your time and consideration.

Reviewer 2 Report

Comments and Suggestions for Authors

Thank you for your efforts! In my opinion, before the paper can be published mayor improvements need to be completed. at the moment line by line suggestions are not useful- 

most important: what is your blank? the formulation without fat? or is there an "inert" ingredient? otherwise the ratios of the other ingredients are different in the blank.
 Was the flour always the same batch? What was the batch size of the doughs? How many doughs did you prepare from each formulation? 

statistics: why didn't you perform a 2-factor analysis (F0 against F2 and all the formulation variations)

Figures and Tables are not standing on their own by now. Explain abbreviations again. Fig. 2 and 4 are not clearly displayed. different scales, no variations, no statistics. so can't evaluate the results.  

in M&M section: devices are missing thoroughly
also specific information on dough mixing (impact, time), conditions during preparation

Intro: needs more specific values, like fat levels

results: stopped reading, as figures can't be seen in detail and statistics are poor

Author Response

(The authors gave the same response as above.)

Round 2

Reviewer 1 Report

Comments and Suggestions for Authors

1. The study only includes two cycles (F0 and F2), which do not adequately represent actual frozen storage conditions. More cycles should be evaluated to observe cumulative effects.
2. No sensory analysis has been performed. It is essential to incorporate a consumer panel evaluation to link instrumental results with the perception of the final product.
3. Although the use of FTIR is mentioned, no figures with spectra or deconvolution are included. It is mandatory to show the representative spectrum and the leading bands, and to explain how the secondary structures were calculated.
4. It was recommended to apply DSC, XRD, colorimetry, Aw, or CLSM. Only FTIR was partially incorporated. It is requested that the other analyses supporting the structural interpretation be included.
5. Additional correlations should be included, and multivariate analysis (PCA or Pearson correlation) should be applied.

Author Response

We sincerely appreciate your valuable feedback and suggestions, and have addressed each point individually. Please see the attachment.
